# Influences of atmospherics on customer satisfaction and behavioural intentions in the restaurant industry: Evidence from an emerging economy

**Mananage Shanika Hansini Rathnasiri**[1]*, **Pawan Kumar**[2], **Bindu Aggarwal**[3], **Kiran Nair**[4], **Narayanage Jayantha Dewasiri**[5,6]

**1** Department of Marketing Management, Faculty of Management Studies, Sabaragamuwa University of Sri Lanka, Belihuloya, Sri Lanka, **2** Mittal School of Business, Lovely Professional University, Phagwara, India, **3** University School of Business, Chandigarh University, Ludhiana, India, **4** Abu Dhabi University, Abu Dhabi, UAE, **5** Department of Accountancy and Finance, Faculty of Management Studies, Sabaragamuwa University of Sri Lanka, Belihuloya, Sri Lanka , **6** Qasim Ibrahim School of Business, Villa College, Malé, The Maldives

* shanika@mgt.sab.ac.lk

## Abstract

The emergence of the food service industry is among the most promising sectors that open many great opportunities for its development. Restaurants are also a part of the world's economy. This is reflected in the increasing demand for high-quality, upscale dining experiences. The restaurant industry is well-positioned for future growth. However, few researchers have explored the impact of restaurant attributes on overall customer satisfaction and behavioural intentions in the context of fine dining restaurants. This study investigates how restaurant attributes influence overall satisfaction and behavioural intentions in an emerging economy such as India. To explore this relationship, we adopted the Stimulus-Organism-Response (SOR) theory proposed by Mehrabian and Russell (1974) [1]. The results show that restaurant attributes can lead to higher customer satisfaction rates, benefiting the industry. Restaurant attributes also play an essential role in creating a positive customer dining experience. Restaurant attributes help provide thoughtful customer service, such as assisting with menu items, remembering customers' names, or providing helpful suggestions, which can also show customers that they are valued.

## 1. Introduction

In the highly competitive food service industry, restaurants in emerging economies face the challenge of creating memorable dining experiences that differentiate themselves and foster customer loyalty. Restaurant appearance, which includes the physical environment, atmosphere, and overall aesthetics, is essential in shaping customers' perceptions, emotions, and preferences [1,2]. Restaurants have become an integral part of society, providing people a place to gather, share meals and create memories. For many, restaurants are a place to relax, enjoy a meal and create lasting memories with family and

**Data availability statement:** This manuscript's data is publicly available via the Sabaragamuwa University of Sri Lanka online repository. Data can be found at the following URL: http://repo.lib.sab.ac.lk:8080/xmlui/handle/susl/4421

**Funding:** The author(s) received no specific funding for this work.

**Competing interests:** The authors have declared that no competing interests exist.

friends. According to the National Restaurant Association of India, India Food Services Report, 2024, the Indian food industry is estimated to be Rs 5,69,487 crore for FY24. It is expected to grow to 7,76,511 crores by FY28, achieving a CAGR of 8.1 per cent, and it will grow with a CAGR of 13.2% [2]. These figures highlight the importance of the restaurant industry in an emerging economy, India and demonstrate its potential for further growth.

Fine Dining restaurants offer a unique and luxurious experience that draws in customers from all walks of life. Whether they are looking for a special occasion, a romantic dinner, or simply a break from the every day, the lure of a fine dining restaurant can be irresistible. The popularity of these establishments is often attributed to their reputation for quality, ambience, and service. This reflects the increasing demand for high-quality, upscale dining experiences. Overall, the growth of the fine dining sector is a positive sign for the restaurant industry. With the number of fine dining establishments and rising spending, this restaurant industry segment is well-positioned for future growth [3].

Fine dining restaurants have a certain level of sophistication associated with them. According to The Spruce Eats, these restaurants usually offer a more formal atmosphere and often have a dress code that patrons must follow. Additionally, they typically have an extensive menu of high-quality dishes prepared with fresh, gourmet ingredients. Moreover, they are known for their attentive, knowledgeable servers and attention to detail regarding presentation. Lastly, fine dining restaurants usually have a higher price point than other restaurants, with meals typically costing more than $30 [3].

The employees of these restaurants are highly competent in preparing food using the best supplies, equipment, and other materials required for business [4]. Such standards of hospitality cannot be compared with anything and exceed the degree of the best-star hotels. Divan, the environment, the music background, and the subdued lighting make the dinners comfortable, luxurious, and lavish. The glassware, crockery, cutlery, and other requisite items used on the serving table are of the best quality [5]. CEOs and head chefs of these restaurants are creative and innovative and have good experience in managing and preparing the meals on the restaurants' menus. In general, highly formal settings, like fine dining restaurants, have certain rules, such as one having to dress appropriately. As a result of formulating competitive strategies to meet the expectations of their clientele, most fine-dining restaurants provide a la carte services.

Therefore, it can be concluded that fine dining restaurants offer their guests an experience rather than an edible product. The current study seeks to fill two research gaps in the literature on fine dining restaurants. First, it examines the role of restaurant attributes in increasing the overall satisfaction of customers and their intentions to re-visit fine dining restaurants, focusing on India, which is among the fastest-growing emerging economies. In many studies related to understanding customer psychology, the theory suggested by Mehrabian & Russel (1974) [1], S-O-R, has been put into practice many times due to its immense relevance with the field of study; it has been put into use. For this purpose, the concept proposed by Mehrabian and Russell (1974) [1] was adopted, which states that many external factors, here, restaurant attributes, act as stimuli and influence the internal state of the person, here, overall customer satisfaction which refers to the organism, which leads to the production of specific behavioural responses, here, behavioural intentions. Second, this study explores the intervening effect of the overall satisfaction of customers between restaurant attributes and intention to return to the same restaurant with particular reference to fine dining restaurants. Furthermore, by considering restaurant features, this study contributes to previous research and paves the way for the fine dining restaurant industry.

## 2. Literature review and research hypotheses

### 2.1. Restaurant attributes

Recent studies have explored the impact of restaurant atmospherics on acceptable dining satisfaction. Specifically, they have found that the atmosphere of a restaurant is an essential factor influencing the enjoyment of a dining experience. In one study, researchers observed that customers in a fine dining restaurant were more likely to value the restaurant atmosphere when perceived as more upscale and luxurious [6]. Moreover, the presence of pleasant lighting, pleasant music, and a pleasant fragrance also contributed to higher levels of satisfaction among customers. The literature has also highlighted the importance of service quality in acceptable dining satisfaction. Researchers have found that customers were likelier to value the restaurant atmosphere when the service was warm and friendly, the waiter staff was knowledgeable and attentive, and the overall food quality was superb. Additionally, the service's personalisation level significantly impacted customer satisfaction [7]. Therefore, restaurant owners need to be aware of the importance of atmospherics to ensure customers have a positive experience.

Recent research has shown that restaurant atmospherics can significantly impact customer satisfaction and motivation to purchase food and beverages. Specifically, the study found that customers who experienced pleasant atmospherics were more likely to rate the restaurant higher and be more motivated to buy food and drink [8]. Another study examined the effects of restaurant atmospherics on customer emotions. The results showed that pleasant atmospherics positively impacted customers' emotional states, with customers feeling more relaxed and happier in a pleasant atmosphere. Further, the study found that customers were more likely to return to a restaurant with a pleasant atmosphere [9]. Finally, a study explored the influence of restaurant atmospherics on customer loyalty. The results showed that customers who experienced lovely atmospherics were more likely to return to the restaurant, indicating increased loyalty. In addition, the study found that customers who experienced pleasant atmospherics were more likely to recommend the restaurant to others. Overall, restaurant atmospherics can significantly impact customers' experiences and behaviour. Specifically, melodic atmospherics can lead to greater customer satisfaction and purchase motivation, improved emotional states, and increased customer loyalty [10]. As such, restaurant owners must create a pleasant and inviting atmosphere for their customers.

### 2.2. Spatial configuration-related attributes in fine dining restaurants

Spatial Configuration's role in fine dining restaurants has been an area of increasing interest in recent years. Studies have sought to understand the motivations and values that drive customers' decisions when selecting a dining experience. Research has shown that spatial configuration motivations are essential in choosing fine-dining restaurants. Customers are more likely to select restaurants perceived as "Configuration significant", meaning they are associated with a particular ambience or cuisine. This includes restaurants that offer traditional and regional dishes and establishments that feature exotic cuisines worldwide. Customers are also often drawn to restaurants that provide a unique atmosphere, such as those with a particular theme or décor [interior design], and those that feature live music, fragrances or other forms of entertainment. In addition, researchers have found that Configuration can be further divided into two distinct categories: Configuration authenticity and Configuration novelty [11,12]. Configuration authenticity refers to the extent to which customers perceive the restaurant as an "authentic" representation of a particular ambience. In contrast, Configuration novelty describes the degree to which customers believe the restaurant offers something new and exciting fine dining experience. Overall, research

has demonstrated that Configuration, including tableware and crockery settings, colour combination schemes and seating arrangements, play an important role in fine dining restaurant selection. Customers are more likely to select restaurants that offer an authentic and novel cultural experience. Understanding these motivations can help restaurant owners and managers better tailor their offerings and create an experience that appeals to their target customers [12].

## 2.3. Sanitation-related attributes in fine dining restaurants

Sanitation is an essential issue for restaurant patrons. A literature review shows that several factors can affect how patrons perceive the sanitation standards of a restaurant. First, the restaurant's physical environment is vital for patrons' appreciation of sanitation. For example, a study of restaurant patrons in the United Kingdom found that the cleanliness of the restaurant environment and the lack of pests were essential factors in determining the sanitation standards of a restaurant [13]. In addition, the Configuration and design of the restaurant can also influence how patrons perceive the cleanliness of a restaurant, as they can determine how easily customers can observe the establishment's cleanliness [14]. Second, the restaurant's sanitation practices can also affect how patrons perceive the restaurant's sanitation standards. A study of restaurant patrons in the United States found that customers were more likely to appreciate the sanitation standards of a restaurant if they felt that the establishment was taking reasonable steps to ensure food safety, such as regularly washing kitchen surfaces and having local health inspections [15]. Third, the restaurant staff's behaviour can impact sanitation-related appreciation. A study in France found that patrons were more likely to appreciate the sanitation standards of a restaurant if they felt that the staff was taking an active role in ensuring cleanliness, such as wearing gloves and masks while handling food [16]. Overall, this literature review suggests that several factors can influence how patrons appreciate the sanitation standards of a restaurant. The physical environment, sanitation practices, and behaviour of restaurant staff all play a role in how customers perceive the cleanliness of a restaurant. Restaurants should consider these factors to ensure that patrons appreciate their sanitation standards.

## 2.4. Music-related attributes in fine dining restaurants

Fine dining restaurants have long been associated with appreciating soothing and delightful music. Music can create an atmosphere of elegance, sophistication, and relaxation, all desirable qualities in a restaurant setting. Studies examining the impact of music on diners' experiences in fine-dining restaurants have been conducted since the 1960s; they have consistently found that music significantly affects overall customer satisfaction. One study found that classical music had a significant influence on customer perceptions of the restaurant environment and the quality of the food. The study found that customers were more likely to rate the restaurant, food, and service more highly when classical music was played [17]. Similarly, another study found that the presence of classical, jazz, or soft rock music in a restaurant led to higher customer ratings of the restaurant overall [18]. Furthermore, it was also found that the type of music played in a restaurant directly impacted customer satisfaction; customers were more satisfied with the restaurant when classical music was playing compared to when jazz or soft rock music was played. The findings of these studies suggest that music-related appreciation in fine-dining restaurants can positively affect customer satisfaction. By playing music appropriate for the type of restaurant, owners can create an atmosphere of sophistication and relaxation that will help customers feel more comfortable and satisfied with their dining experience [18].

### 2.5. Menu-related attributes in fine dining restaurants

Menu-related items in fine dining restaurants are essential to creating an enjoyable and memorable dining experience. Recent research suggests that a menu's design, presentation, and content can significantly impact a diner's perceptions of the overall experience. In a research study, participants were exposed to different menu designs and asked to rate the restaurant experience they believed they had. Results showed that diners were more likely to order the restaurant experience positively when menus were designed with an attractive, organised, and modern design. Furthermore, menu items with vivid descriptions and imagery were associated with higher ratings [19]. Menu descriptions played a vital impact on customer satisfaction. The menu was categorised by the type of information they provided, such as nutrition information, ingredient lists, and allergen warnings. Results showed that menus that provided more detailed information were associated with a higher appreciation for the restaurant experience. This was especially true for menus that provided nutrition information, which was the most influential factor in generating appreciation. These studies demonstrate that visually appealing menus with organised and modern designs, detailed information, vivid descriptions and imagery are more likely to create positive restaurant experiences. Therefore, restaurants need to focus on the design and content of their menus to provide customers with an enjoyable and memorable dining experience [20].

## 3. Operational Definitions of Constructs

### 3.1. Spatial Configuration-related attributes

A restaurant's Configuration design refers to the physical arrangement of various elements such as tables, chairs, bar area, kitchen, reception, and other features. It involves planning and organising the restaurant's physical space to optimise functionality, improve traffic flow, and create an enjoyable dining experience for customers [11,12].

### 3.2. Sanitation-related attributes

Sanitation can be described as cleanliness and other measures to avoid spreading diseases. I think that the three aspects, namely, food, climate, and service, which customers know from a sanitation perspective, are proof of the restaurant's sanitation. Given that consumers primarily experience the front of the house in restaurants, it is expected that the actual physical environment in which consumers physically interact, including the dining room, tables, servers, and toilets, will have a significant influence on customer hygiene preferences, moods arising from interaction with the restaurant environment, and consumers' behaviour [13,14].

### 3.3. Music-related attributes

Music reduces stress and relieves anxiety. Backdrop music, tempo, and pattern of music, among others, are some of the musical features. Studies on restaurant music have highlighted that this factor influences food choices and eating behaviour and causes clients to take longer waiting and eating times, order more food, and more costly food [17,18].

### 3.4. Menu-related attributes

As stated above, a menu is a tool through which a restaurant conveys information relating to food and services to clients. This can be done through menu design and the description of food items on the menu card [19, 20]. Menu design produces an aesthetically pleasing menu card that encompasses details and catches consumers' attention to certain products that the food service entity wants to sell [21, 22].

### 3.5. Customer satisfaction

Customer satisfaction may be defined as a customer's attitude emerging from the perceived quality of a product or service he has bought or used and the price tag that he had to pay for that product or service [23]. Customer satisfaction is the evaluative response of a customer towards a service provider [24]. Customers' expectations have specific predictors, such as perceived service, price, food quality and restaurant atmosphere [25,26].

### 3.6. Behavioural intentions

Behavioural intentions are defined as a person's conscious action plan, which involves directed effort towards executing specific behaviours and intentions from personal perceptions and norms. Therefore, involvement in a particular behaviour equals behavioural intentions [27]. Behavioural intentions are the driving force behind random-struck conduct and are directly associated with the customer [28] (Fig 1).

As a result, the following hypotheses are advanced.

H1: Restaurant atmospherics positively influence (i)overall customer satisfaction & (ii) behavioural intentions.

H1(a): Spatial Configuration-related attributes positively influence (i)overall customer satisfaction and (ii) behavioural intentions.

HI (b): Sanitation-related attributes positively influence (i) overall customer satisfaction and (ii) behavioural intentions.

HI (c): Music-related attributes positively influence (i) overall customer satisfaction and (ii) behavioural intentions.

HI (d): Menu-related attributes positively influence (i) overall customer satisfaction and (ii) behavioural intentions.

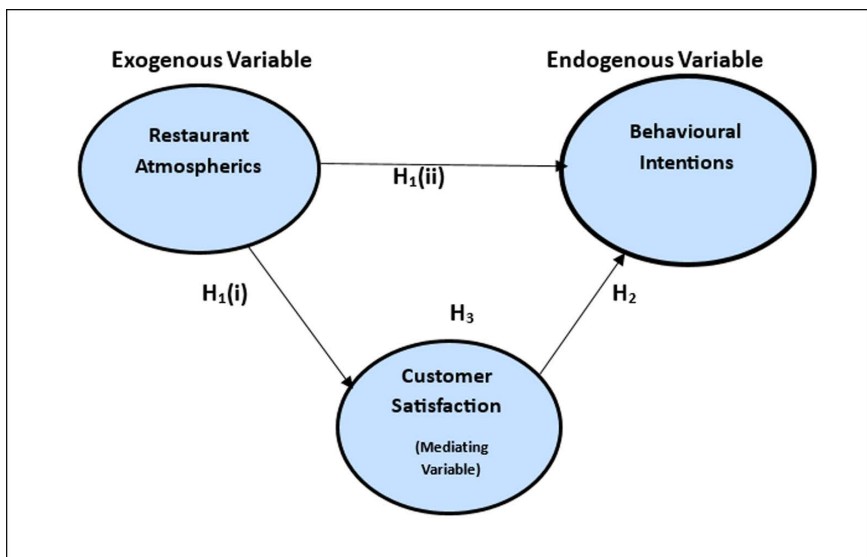

**Fig 1. Proposed Model of Restaurant Atmospherics, Overall Customer Satisfaction, and Behavioural Intentions.**

### 3.7.  Overall customer satisfaction for fine dining restaurants

Recent research has shown that restaurant atmospherics play an essential role in overall customer satisfaction to appreciate the dining restaurant attributes of establishments. Specifically, researchers have found that customers are more likely to enjoy a restaurant if it is well-designed, has a pleasant atmosphere, and offers a high-quality dining experience [29, 30]. In a study by [31], the authors investigated how restaurant atmospherics impact customers' experience. The authors found that customers respond positively to restaurants with good design, comfortable seating, and an inviting ambience. Furthermore, the authors concluded that customers are likelier to appreciate a restaurant with an aesthetically pleasing atmosphere [32]. Customers have more potential to enjoy a restaurant's attributes if it has a pleasant atmosphere, good service, and a high-quality dining experience. The authors concluded that customers are more likely to revisit a restaurant if they enjoy the atmosphere [33]. Customers have more potential to appreciate a restaurant with a pleasant atmosphere, good food, and a welcoming environment. Overall, the evidence suggests that restaurant atmospherics are essential in motivating customers to appreciate fine dining establishments [34]. Customers are more likely to enjoy a restaurant if it is well-designed, has a pleasant atmosphere, and offers a high-quality dining experience. Thus, restaurant owners should create an aesthetically pleasing atmosphere conducive to customer enjoyment [35].

Customer satisfaction is essential for fine dining, and restaurant attributes appreciate appreciation. Customers often prefer to dine in places where they feel appreciated and valued. A study by the National Restaurant Association found that customers who experienced "excellent customer appreciation" were more likely to recommend the restaurant to friends and family and return for future meals. Studies have also found that customers are more likely to appreciate a restaurant's efforts if they know them [35].

**Hypothesis 2**: Overall customer satisfaction influences customers' behavioural intentions positively.

### 3.8.  Behavioral Intentions

The relationship between customer appreciation for fine dining restaurants and behavioural intentions has been studied in several studies. Higher levels of appreciation for fine dining restaurants were more likely to engage in repeat visits and recommend the restaurant to others. This suggests that customers have a positive attitude towards fine dining restaurants and are willing to patronise them in the future [36]. A study found that customers with higher appreciation for fine dining restaurants were likelier to leave a positive review, thus contributing to the restaurant's reputation. Furthermore, the study showed that customers with higher levels of appreciation were also more willing to spend more money on their dining experience. These findings suggest that customers appreciate the high quality of food and service offered by fine dining restaurants and are willing to pay a premium for it [37]. Recently, a study examined the relationship between customer appreciation for fine-dining restaurants and their loyalty intentions. The study found that customers with higher appreciation for the restaurant were likelier to become loyal. This suggests that customers appreciate the quality of the fine dining restaurants' food, service, and atmosphere and are willing to pay more for the experience [38,39,40]. Overall, the research suggests that customers appreciate the high quality of food and service offered by fine dining restaurants and are willing to pay a premium for the experience. Furthermore, customers with a higher level of appreciation for the restaurant are more likely to become loyal and recommend the restaurant to others. Therefore, it is essential for fine dining restaurants to offer excellent food and service to their customers to foster customer loyalty and appreciation [41].

In recent years, the dining experience at fine dining restaurants has become increasingly important as consumers seek a unique and memorable experience. This has led to an increased focus on understanding consumers' behaviours in such restaurants and the factors that influence their decision-making process. One such factor is behavioural intentions, an individual's decision to engage in specific behaviour. This literature review explores existing research on customers' behavioural intentions in fine-dining restaurants [42]. A study examined the effect of restaurant characteristics on customers' behavioural intentions in fine dining restaurants. The study found that both the service quality and the ambience of the restaurant had a positive effect on customers' behavioural intentions. Specifically, customers were more likely to have a positive behavioural intention when the restaurant provided excellent service and a pleasant atmosphere [43]. Another study analysed the role of customer satisfaction and perceived value in influencing customers' behavioural intentions in fine dining restaurants. The study showed that customer satisfaction significantly affected behavioural intentions, while perceived value had no significant impact. The authors concluded that providing an excellent dining experience influences customers' behavioural intentions. They finally explored the role of pricing in influencing customers' behavioural intentions in fine dining restaurants. The results showed that customers' willingness to pay more for the same meal positively affected their behavioural intentions. The authors suggested that offering a range of prices can help to increase customers' willingness to pay, thus increasing their behavioural intentions. Overall, the literature on customers' behavioural intentions in fine-dining restaurants shows that several factors can influence their decision-making [44]. These factors include restaurant characteristics, customer satisfaction, perceived value, and price. As such, restaurant owners need to understand the importance of these factors to provide an excellent dining experience and increase customers' behavioural intentions. These findings imply that appreciation is a crucial predictor of behavioural intentions in the premium restaurant sector, which gives rise to the following hypotheses:

Hypothesis 3: Overall customer satisfaction mediates the relationship between restaurant atmospherics and behavioural intentions.

## 4. Methodology

### 4.1. Research instrument

When it comes to the process of obtaining valuable and relative information from the customers, then it is crucial to create a questionnaire. The completion of a questionnaire involves ingenuity, and one cannot just wake up one morning and create a questionnaire and administer it. In this regard, a questionnaire was constructed and consisted of statements about restaurant atmospherics (spatial configuration, sanitation, music & menu), overall customer satisfaction, and behavioural intention. The attributes for the dimensions as mentioned earlier have been taken from the scales developed by earlier researchers and were modified: Tangible Service Attributes [45], SERVICESCAPE [46], DINESCAPE [47], SERVQUAL [48], DINESERV [49], TANGSERV [50] & DINEX [51] (Table 1).

### 4.2. Data collection

The second part of the questionnaire is aimed at collecting data concerning the assessment of overall customer satisfaction [52,53] as well as behavioural intentions [49,54,55]. The last part describes the socio-demographic characteristics of the respondents. This research was performed from August to December of the year 2023 in an elite restaurant that specialises in Indian cuisine, situated in New Delhi, India. The population sample was chosen by adopting the systematic sampling technique where the researcher interviewed every second table (the

**Table 1. Research Instrument.**

| Variable | Indicators | Source |
|---|---|---|
| Spatial Configuration-related attributes | B1. I like the interior design of the restaurant. | 45,46,47,48 |
| | B2. I like the natural fragrance in the restaurant. | |
| | B3. I like the tableware and pottery used in the restaurant. | |
| | B4. I like the colour scheme of the restaurant. | |
| | B5. I like the restaurant's seating arrangement. | |
| Sanitation-related attributes | C2. I like the cleanliness of the washrooms. | 45,46,47,48 |
| | C3. I like the availability of toilet paper in the washroom. | |
| | C4. I like the availability of water, towels, and soap in the washroom. | |
| | C5. I like that the dustbins were properly placed and covered. | |
| | C6. I like the cleanliness of the washbasin, wall mirror, and floor of the washroom. | |
| | C7. I like that the restaurant staff was neatly dressed. | |
| Music-related attributes | D1. I like the background music played. | 45,46,47,48 |
| | D2. I like the apt fit of the background music with the restaurant's image. | |
| | D3. I like that the music played was familiar to me. | |
| | D4. I like the tenderness and volume of the music. | |
| | D5. I agree that the music made dining more fun. | |
| | D6. I agree that music extended my stay at the restaurant. | |
| Menu-related attributes | M1. I like the attractiveness of the menu card. | 49,50,51 |
| | M2. I like the easy readability of the menu card. | |
| | M3. I like the variety of food choices available. | |
| | M4. I like locating a food item on the menu card, which is easy. | |
| | M5. I like the price tags. | |
| Overall Customer Satisfaction | F1 I am delighted with the overall experience at this restaurant. | 52,53 |
| | F2 As a whole, I am happy with the restaurant. | |
| | F3 I believe I did the right thing to visit the restaurant. | |
| | F4 I find the restaurant to be an overall enjoyable experience. | |
| Behavioural intentions | G1. I want to return to this place for another meal. | 49,54,55 |
| | G2. I want to share good vibes about this eatery with others. | |
| | G3. You will dine at this restaurant in future. | |

respondents). Overall, 500 questionnaires were made available in relation to this research. The final sample mean reached 440 people. The demographic profile of the respondents is stated in Table 2.

## 5. Data analysis and presentation

The research model in this study is estimated and evaluated using Structural Equation Modelling (SEM). The analysis is done in compliance with the prescribed guidelines, procedures, and critical values, as explained elsewhere [56,57]. All the constructs in our model (Fig 2) assess reflective measurement models of indicator reliability, internal consistencies, reliability, convergent validity, and discriminant validity.

### 5.1. Convergent Validity

The assumption of convergent validity (CV) relies on three key prerequisites. To ensure convergence on a construct, all standardised loading estimates should be statistically significant

Table 2. Demographic Profile.

| Demographic Variables | Category | Frequency | Percentage (%) |
|---|---|---|---|
| Age | 18–30 | 170 | 38.6 |
| | 31–40 | 130 | 29.5 |
| | 41–50 | 73 | 16.6 |
| | 51–60 | 40 | 9.1 |
| | 61&above<br>Total | 27<br>440 | 6.1 |
| Gender | Male | 205 | 46.6 |
| | Female<br>Total | 235<br>440 | 53.4 |
| Marital Status | Married | 244 | 55.5 |
| | Unmarried<br>Total | 196<br>440 | 44.5 |
| Occupation | Student | 120 | 27.3 |
| | Self-Employed | 61 | 13.9 |
| | Public Sector | 50 | 11.4 |
| | Private Sector | 102 | 23.2 |
| | Professional | 62 | 14.1 |
| | Another<br>Total | 45<br>440 | 10.2 |

(value of 0.50 or higher) [58]. The second requirement is the average variance extracted (AVE) of 0.50 or greater as proof of appropriate convergence. Third, the criterion is construct reliability (CR) of 0.7 or more, demonstrating high reliability. All of these criteria indicate that the CV assumption is not violated in this investigation [58] (Table 3).

## 5.2. Discriminant Validity

When the correlation between exogenous variables exceeds the square root of average variance extraction (AVE), the discriminant validity condition is violated [59]. To calculate discriminant validity, master validity choices were employed, including standard regression weights and correlation across all constructs. Table 4 shows the output from AMOS 24, which confirmed that the square root of the AVE value (diagonal value in bold) is larger than the inter-construct correlation coefficient. Thus, the assumption of discriminant validity isn't violated. Once the measurement model was established, the next step was to evaluate the structural model to examine the stated hypotheses.

## 5.3. Structural model (Direct Effects)

This study model included three variables: restaurant atmospherics (exogenous), overall customer satisfaction (exogenous), and behavioural intentions (endogenous). This section explains the direct relationship between these three primary variables.

 **5.3.1. Relationship between restaurant atmospherics, overall customer satisfaction, and behavioral intentions.** The study aimed to examine how restaurant atmospherics and overall customer satisfaction influence. H(1) consists of three fundamental hypotheses. As shown in Fig 3 & Table 5, the structural model findings validated hypothesis H1(i), which states that there is a positive link between restaurant atmospherics and customer satisfaction, with path coefficient value for Spatial configuration-related attributes ($\beta$ = -.05, critical value $t$ -2.114, $p < 0.05$), sanitation-related attributes ($\beta$ = 0.337, critical value $t$ 8.566, $p = 0.000$),

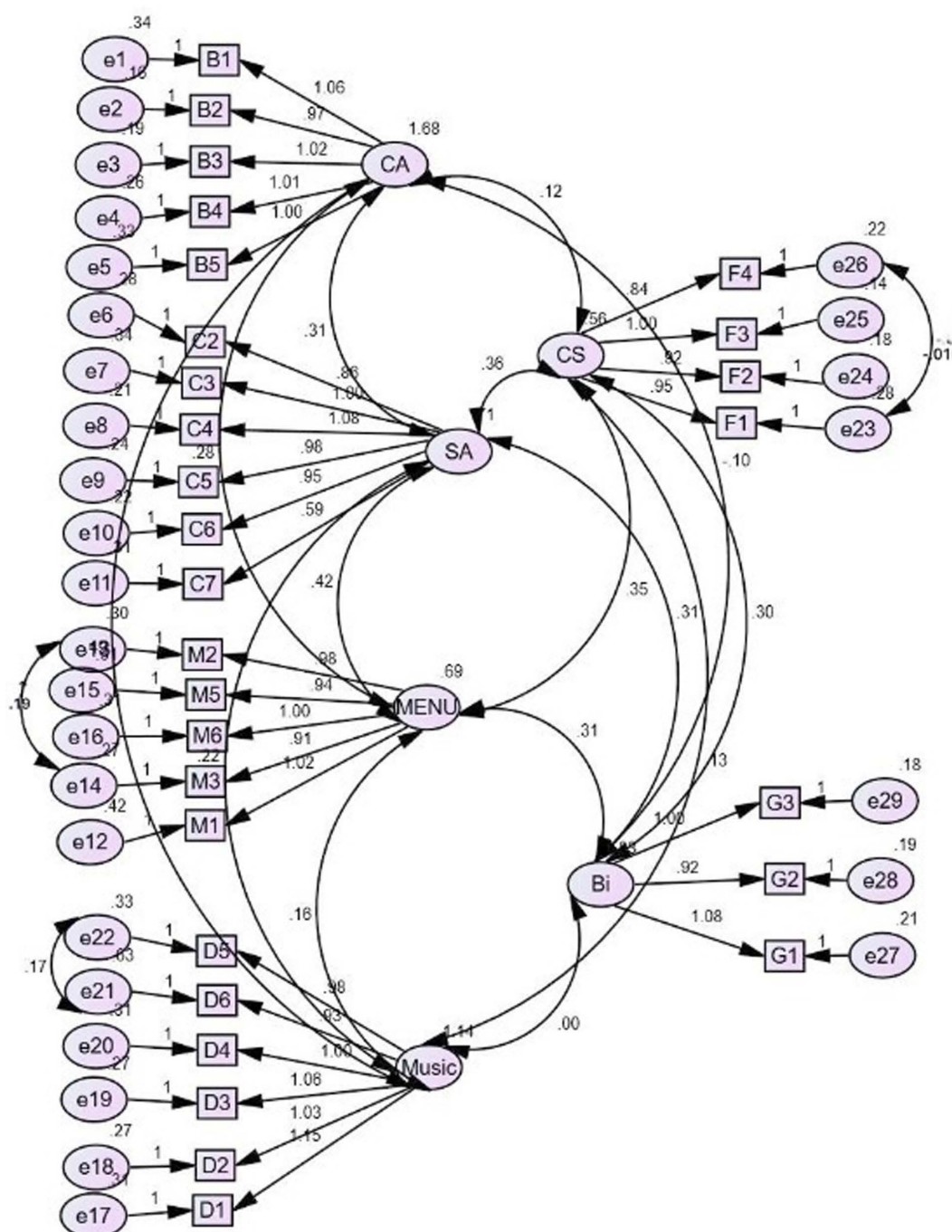

**Fig 2. Overall Measurement Model of Restaurant Attributes, Overall Customer Satisfaction, and Behavioral Intentions.**

music-related attributes (β = 0.033, critical value $t$ 1.165, $p$ < 0.05), and menu-related attributes (β = 0.322, critical value $t$ 8.015, $p$ = 0.000) except for the music-related attributes. This study also finds that except music-related attributes, all other restaurant attributes (Spatial Configuration (β = -.147, critical value $t$ -4.745, p < 0.05), sanitation (β = .245, critical

**Table 3. Reliability and Validity of the Overall Measurement Model (Convergent Validity).**

| Latent Variable | Items | Standardised loadings | Composite Reliability | Cronbach Alpha | AVE |
|---|---|---|---|---|---|
| Spatial Configuration-related | B1 | .920 | 0.971 | 0.823 | 0.871 |
| | B2 | .952 | | | |
| | B3 | .950 | | | |
| | B4 | .932 | | | |
| | B5 | .913 | | | |
| Sanitation-related | C2 | .808 | 0.925 | 0.885 | 0.673 |
| | C3 | .821 | | | |
| | C4 | .890 | | | |
| | C5 | .858 | | | |
| | C6 | .862 | | | |
| | C7 | .664 | | | |
| Music-related | D1 | .912 | 0.953 | 0.890 | 0.773 |
| | D2 | .906 | | | |
| | D3 | .910 | | | |
| | D4 | .887 | | | |
| | D5 | .875 | | | |
| | D6 | .780 | | | |
| Menu-related | M1 | .798 | 0.855 | 0.892 | 0.671 |
| | M2 | .838 | | | |
| | M3 | .829 | | | |
| | M5 | .815 | | | |
| | M6 | .833 | | | |
| Overall Customer Satisfaction | F1 | .801 | 0.904 | 0.901 | 0.703 |
| | F2 | .852 | | | |
| | F3 | .896 | | | |
| | F4 | .800 | | | |
| Behavioural Intentions | G1 | .905 | 0.926 | 0.895 | 0.808 |
| | G2 | .887 | | | |
| | G3 | .904 | | | |

**Table 4. Discriminant Validity Analysis of the Overall Measurement Model.**

| | Spatial Configuration | Sanitation | Music | Menu | Overall Customer Satisfaction | Behavioural Intentions |
|---|---|---|---|---|---|---|
| Spatial Configuration | **0.933** | | | | | |
| Sanitation | .270** | **0.821** | | | | |
| Music | .337** | .239** | **0.879** | | | |
| Menu | .236** | .551** | .176** | **0.819** | | |
| Overall Customer Satisfaction | .121* | .534** | .161** | .523** | **0.838** | |
| Behavioural Intentions | −.084 | .377** | .010 | .380** | .402** | **0.899** |

*The diagonal values are the square root of each construct's 'Average Variance Extracted'(AVE); below the diagonal are the correlations.

**p < 0.01(inter-construct correlation value)

value *t* 4.403, p < 0.05), and menu (β = .252, critical value *t* 4.432, p < 0.05) have significant positive effects on behavioural intentions [H1(ii)]. Hypothesis H2: customer satisfaction is favourably correlated with behavioural intentions path coefficient value β = 0.263, critical value *t* 3.439, *p* = 0.000, as illustrated in Fig 3 (Table 6).

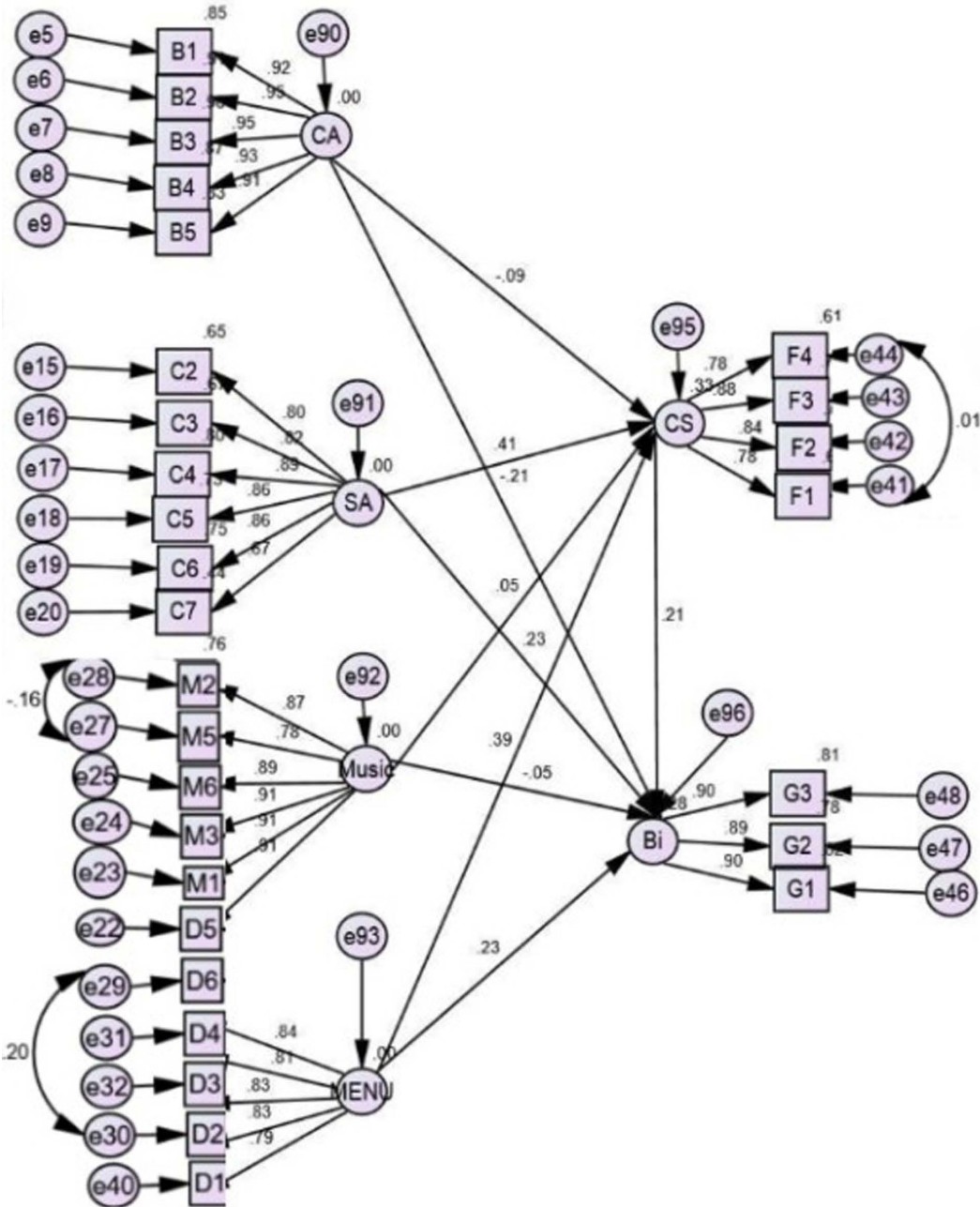

**Fig 3. Hypothesised Structural Equation Model with Standard Estimates.** Note: CA=Spatial Configuration-related attributes, SA=Sanitation-related attributes, M=Menu-related attributes, D=Music-related attributes, CS=Overall Customer Satisfaction, and BI=Behavioral Intentions.

**Table 5. Standard Regression Weights for the Structural Equation Model.**

|  | Relationships |  | Estimate (β) | S.E. | C.R(*t value*) | P |
|---|---|---|---|---|---|---|
| CS | <--- | SA | .337 | .039 | 8.566 | *** |
| CS | <--- | MENU | .322 | .040 | 8.015 | *** |
| CS | <--- | Music | .033 | .029 | 1.165 | .244 |
| CS | <--- | CA | −.050 | .023 | −2.114 | .035 |
| Bi | <--- | CS | .263 | .077 | 3.439 | *** |
| Bi | <--- | Music | −.041 | .037 | −1.099 | .272 |
| Bi | <--- | MENU | .252 | .057 | 4.432 | *** |
| Bi | <--- | SA | .245 | .056 | 4.403 | *** |
| Bi | <--- | CA | −.147 | .031 | −4.745 | *** |

***Significant at 0.001 level.

**Table 6. Structural Model Direct Effects.**

| Hypothesised path | Standardised path coefficients | t-value | p-value | Results |
|---|---|---|---|---|
| H1a(i) | −.050 | −2.114 | .035 | supported |
| H1b(i) | .337 | 8.566 | *** | supported |
| H1c(i) | .033 | 1.165 | .244 | not supported |
| H1d(i) | .322 | 8.015 | *** | supported |
| H1a(ii | −.147 | −4.745 | *** | supported |
| H1b(ii) | .245 | 4.403 | *** | supported |
| H1c(ii) | −.041 | −1.099 | .272 | not supported |
| H1d(ii | .252 | 4.432 | *** | supported |
| H2 | .263 | 3.439 | *** | supported |

***Inter-construct correlation value.

**Table 7. Direct & Indirect Effects- Two-Tailed Significance (BC).**

| Effect | Spatial Configuration | Sanitation | Music | Menu | Overall Customer Satisfaction | Behavioural Intentions |
|---|---|---|---|---|---|---|
| Overall customer satisfaction | … | … | … | … | … | … |
| Behavioural Intentions (Direct Effect) | 0.001 | 0.001 | 0.324 | .001 | 0.001 | … |
| Behavioural Intentions (Indirect Effect) | 0.04 | 0.001 | 0.355 | .001 | … | … |

Source: Authors' own

**5.3.2. Sem-Mediation through the boots trap approach.** The relationship between restaurant atmospherics and behavioural intentions, with the mediating role of customer satisfaction, was examined. The Bootstrapping method on AMOS software was employed to analyse the mediation effect. Table 7 shows that overall customer satisfaction partially mediates the relationship between restaurant atmospherics and behavioural intentions, as it has both direct and indirect influences. All the exogenous latent constructs except music significantly directly affect behavioural intentions. Hence, a partial mediating effect of customer satisfaction was found between restaurant atmospherics and behavioural intentions.

## 6. Findings

The Mehrabian and Russell model was employed in this study to determine the relationship between the stimulus, restaurant atmospherics, and the organism's customer satisfaction, with the response being behavioural intentions. According to literature analysis, CRM is an assertion that restaurant atmosphere influences customer satisfaction and subsequent behaviour [60, 25, and 26]. The current study also found that restaurant atmosphere influences consumer satisfaction and behavioural intentions. Customers ranked tablecloths and crockery as the most important atmospheric attributes in terms of satisfaction and behavioural intentions [65,66,67]. As customer satisfaction increased, cultural elements like the wall colour scheme, seating arrangement, and others added value to the restaurant. Remarkably, the least amount of influence was found to be associated with the restaurant's interior design and the aroma or odour that it imparted to its patrons, which in turn affected the patrons' perceived behavioural intentions and degree of satisfaction [61,62,63,64]. Customers rated tidy attire the most out of all the hygienic traits that the restaurant staff considered to be vital. The location of the dustbins and washbasin, the wall mirror, and the cleanliness of the floor were found to be the most significant restroom facility qualities that influenced patron pleasure and behavioural intentions in fine-dining restaurants [65,66,67]. The fact that listening to delicate, quiet music was preferable to loud music was also found to be evidence [68, 69]. The truth is that the current study solely discussed the patrons' favourable opinions of the music that was played in fine dining establishments. Customers did not appear to be inclined to return to the restaurant in order to enjoy the music. Among the other menu features, the delivery of various dishes on the menu card was the most favoured [70, 71, 72]. The study found that overall customer satisfaction partially mediates the association between restaurant features and behavioural intention.

## 7. Discussion and Implications

The purpose of this research is to identify the factors related to restaurant characteristics that contribute to the level of satisfaction and further behavioural patterns among consumers of fine dining restaurants. Therefore, it can be seen that analysing the impact of motivators of fine dining restaurants on customers' satisfaction levels and behaviour intention can be considered an important issue in the restaurant industry. The quality of the food, service, and ambience all impact a customer's experience and can influence their appreciation of the restaurant and their intentions to return. First and foremost, the Spatial Configuration of a restaurant is a significant motivator for customers. The taste, presentation, and overall food quality will significantly influence their enjoyment of the restaurant [10,13]. Additionally, the restaurant's service also positively impacts customer satisfaction. If the staff is friendly, knowledgeable, and attentive, it can make the dining experience more enjoyable and desirable. Finally, the restaurant's atmosphere is also essential for customer satisfaction. Whether in a formal or a casual setting, the restaurant's ambience can significantly influence the customer's final impression [73]. Regarding behavioural intentions, these three attributes are also crucial in influencing customers to return [24].

An excellent dining experience also makes customers repeat customers and even go the extra mile of recommending the restaurant to their friends and family. On the other hand, when the food, service, and environment are poor, customers tend to avoid returning to the restaurant and may not recommend it [74]. In sum, food quality, service quality, and ambience quality are critical services that restaurants must consider in order to ensure customers' behavioural intentions. All these attributes can go a long way in defining a customer's experience and greatly influence the restaurant's fortunes [72].

Another pertinent area that has dominated research on the determinants of restaurant patrons' behaviour has been the mediating variable of customer satisfaction between restaurant attributes as motivators of behavioural intentions for the fine dining restaurant industry. The study has revealed that restaurant attributes like quality of food, service quality, atmosphere, and price can act as stimuli that may cause a behavioural response in the form of restaurant revisit, recommendation of the restaurant to other people and Increased expenditure on the meal. Therefore, it can be identified that customer satisfaction moderates the range of restaurant attributes and impacts behavioural intentions [64].

## 7.1 Implications for practice

The findings of this research have several implications for the hospitality industry. First, they suggest that restaurant managers must consider customer satisfaction when developing strategies to increase loyalty and satisfaction. Customer satisfaction regarding specific attributes of a restaurant can drive positive behavioural intentions. Thus, restaurant managers should focus on providing high-quality, consistent services and creating a pleasant atmosphere to encourage customer satisfaction [55].

Second, these findings suggest that restaurant managers should use customer satisfaction to engage customers and increase loyalty. Customer satisfaction can also be achieved through feedback about the restaurant's attributes, which can help managers identify what aspects of the restaurant are most valued by customers. Focusing on customer satisfaction can help restaurant managers increase customer loyalty, satisfaction, and spending, and thus, should be considered when developing strategies to increase success.

This study considers several factors for premium restaurant managers. To provide an elevated dining experience for customers, fine dining restaurant managers should strive to create an enriched atmosphere. Managers should focus on enhancing the spatial layout by paying particular attention to space utilisation, furniture selection, artwork, décor, etc., as these elements can help create a unique, memorable setting for guests, increasing customer satisfaction and behavioural intentions. By utilising these strategies, managers can create a unique, culturally rich atmosphere for guests, which can help to increase customer satisfaction and behavioural intentions [56].

Menu-enthused elements can be instrumental in this regard. By offering customers the opportunity to customise their meals and create unique experiences, managers can demonstrate their commitment to customer satisfaction [19]. Additionally, offering discounts and incentives to loyal customers can help to increase their loyalty and appreciation for the restaurant. Furthermore, creating a sense of community and connecting with customers through social media can help to foster relationships and create a more positive customer experience [30]. Finally, providing personalised service to customers, including attentive wait staff and personalised suggestions, can help enhance the customer experience and create a lasting impression.

Managers should prioritise sanitation elements to improve customer satisfaction and behavioural intentions in a fine dining restaurant. This includes regularly cleaning and sanitising dining and kitchen areas, providing hand sanitiser at the restaurant's entrance, ensuring that staff wash their hands frequently, and giving single-use condiment containers and disposable cutlery [29]. In addition, managers should encourage staff to wear masks and gloves and to take other safety precautions. Moreover, managers should ensure that all tables, chairs, and surfaces are wiped down between customers and that any buffet-style food is served by staff, not self-serve. Finally, managers should ensure that all staff know the importance of providing customers with a clean and safe dining environment. By taking these steps, managers can

show customers they are taking their health seriously and create a positive experience for them [28].

As a fine dining restaurant manager, there are a few essential implications when incorporating music-enthusing elements into the customer experience. First, selecting music that aligns with the restaurant's ambience and overall aesthetic is necessary. Music should be chosen carefully to create a relaxing and enjoyable atmosphere for customers. Additionally, the volume of the music should be kept at a comfortable level so that it does not interfere with conversations or disrupt the overall dining experience [19]. Finally, monitoring customer reactions and adjusting the music is essential. By doing so, customers feel appreciated, and their behavioural intentions towards the restaurant can be improved. Ultimately, by carefully incorporating music-enthused elements, fine-dining restaurant managers can create a memorable and enjoyable experience for their customers [63].

Fine dining restaurant managers should strive to create an atmosphere of appreciation and respect for their customers. This can be achieved through various methods, including providing excellent customer service, creating a welcoming and friendly atmosphere, and offering special promotions and discounts. Additionally, restaurant managers should work to build relationships with their customers by listening to their feedback and suggestions, responding quickly to their inquiries, and offering personalised experiences. By creating a positive customer experience, restaurant managers can help to improve customer appreciation and behavioural intentions. Additionally, providing customers with incentives such as discounts and loyalty programs can help to strengthen customer loyalty and encourage repeat business. Finally, establishing clear communication and expectations between managers and customers can help to ensure that customers are satisfied with their experiences and are motivated to return in the future. The findings of this study could be of great importance for the fine-dining industry, specifically in the context of an emerging economy like India, as the food service industry is estimated to increase to 125 billion U.S. dollars in 2029 [2]. This estimate accounts for full-service restaurants, quick-service restaurants, cafes and bars and the cloud kitchen segment, paving the way for the restaurant industry to flourish.

## 7.2 Theoretical implications

The Stimulus-Organism-Response (S-O-R) model gives useful information on consumer behaviour in the fine dining industry. This theoretical framework proposes that external stimuli influence an organism's internal processes, resulting in distinct behavioural responses. In emerging nations where the restaurant sector, especially fine dining, is thriving, the S-O-R model is crucial for comprehending the impact of restaurant traits on customer satisfaction and behavioural intentions. The current study contributes to the existing literature. It identifies notable characteristics of fine dining restaurants that, if well addressed, could enhance the entire customer experience and lead to favourable behavioural intentions.

## 8. Limitations and future research directions

There are some drawbacks to this study. First, this study does not consider the role of restaurant staff as a source of customer satisfaction. Motivated and satisfied staff could be able to recognise the level of customer satisfaction and take steps to ensure that they have a memorable experience. Second, the questionnaire was completed by respondents based on their perceptions of Indian fine dining restaurants. Future research could replicate this study in a different cultural environment to strengthen the generalizability of our findings. Finally, future research might investigate the moderating effect of demographics on customer satisfaction and the intent to visit Fine Dining Restaurants. Customer expectations

and opinions are greatly influenced by social media and online evaluations, which replicate modern dining conventions. Future researchers could potentially examine these characteristics' effects on customer satisfaction and behavioural intentions by using the S-O-R model.

## 9. Conclusions

The SOR model offers a significant framework for examining the impact of stimuli on an organism and its subsequent responses [75]. This study postulates perceived values regarding restaurant attributes as motivation variables that lead to satisfaction. Fine dining restaurants can improve customer satisfaction and behavioural intentions by providing a high-quality dining experience, offering personalised services, utilising customer feedback to make meaningful changes, and focusing on creating a comfortable and inviting atmosphere [64]. By doing these things, customers will feel more appreciated and be more likely to return to the restaurant. Additionally, restaurants must build customer relationships by engaging in meaningful conversations and providing excellent customer service. By doing this, customers will be more likely to have positive behavioural intentions towards the restaurant and be more likely to recommend it to others [48]. The empirical findings reveal that satisfaction, embodied by perceived values, is certainly positively and significantly connected to behavioural intentions, reaffirming earlier studies' findings. The study also discovers that appreciation plays a moderating role in the relationship between restaurant motivation and behavioural intentions and the association between restaurant motivation and satisfaction.

## Author contributions

**Conceptualization:** Mananage Shanika Hansini Rathnasiri.

**Data curation:** Mananage Shanika Hansini Rathnasiri.

**Formal analysis:** Mananage Shanika Hansini Rathnasiri, Pawan Kumar.

**Investigation:** Mananage Shanika Hansini Rathnasiri, Kiran Nair, Bindu Aggarwal.

**Methodology:** Mananage Shanika Hansini Rathnasiri, Pawan Kumar, Narayanage Jayantha Dewasiri, Kiran Nair, Bindu Aggarwal.

**Project administration:** Mananage Shanika Hansini Rathnasiri, Pawan Kumar, Narayanage Jayantha Dewasiri, Kiran Nair, Bindu Aggarwal.

**Resources:** Mananage Shanika Hansini Rathnasiri, Pawan Kumar, Narayanage Jayantha Dewasiri, Kiran Nair, Bindu Aggarwal.

**Software:** Mananage Shanika Hansini Rathnasiri, Pawan Kumar, Narayanage Jayantha Dewasiri, Kiran Nair, Bindu Aggarwal.

**Supervision:** Pawan Kumar, Narayanage Jayantha Dewasiri, Kiran Nair, Bindu Aggarwal.

**Validation:** Mananage Shanika Hansini Rathnasiri, Pawan Kumar, Narayanage Jayantha Dewasiri, Kiran Nair, Bindu Aggarwal.

**Visualization:** Mananage Shanika Hansini Rathnasiri, Pawan Kumar, Narayanage Jayantha Dewasiri, Kiran Nair, Bindu Aggarwal.

**Writing – original draft:** Mananage Shanika Hansini Rathnasiri, Pawan Kumar, Narayanage Jayantha Dewasiri, Kiran Nair, Bindu Aggarwal.

**Writing – review & editing:** Mananage Shanika Hansini Rathnasiri, Pawan Kumar, Narayanage Jayantha Dewasiri, Kiran Nair, Bindu Aggarwal.

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
