## [Decision Letter · Decision Letter 0]

23 Jun 2024

PONE-D-24-10577Influences of Atmospherics on Customer Satisfaction and Behavioral Intentions in the Restaurant Industry: Evidence from an Emerging EconomyPLOS ONE

Dear Dr. Rathnasiri,

Thank you for submitting your manuscript to PLOS ONE. After careful consideration, we feel that it has merit but does not fully meet PLOS ONE’s publication criteria as it currently stands. Therefore, we invite you to submit a revised version of the manuscript that addresses the points raised during the review process.

We look forward to receiving your revised manuscript.

Kind regards,

Yasuko Kawahata

Academic Editor

PLOS ONE

Journal Requirements:

2. Thank you for submitting the above manuscript to PLOS ONE. During our internal evaluation of the manuscript, we found significant text overlap between your submission and previous work in the [introduction, conclusion, etc.].

Please revise the manuscript to rephrase the duplicated text, cite your sources, and provide details as to how the current manuscript advances on previous work. Please note that further consideration is dependent on the submission of a manuscript that addresses these concerns about the overlap in text with published work.

[If the overlap is with the authors’ own works: Moreover, upon submission, authors must confirm that the manuscript, or any related manuscript, is not currently under consideration or accepted elsewhere. If related work has been submitted to PLOS ONE or elsewhere, authors must include a copy with the submitted article. Reviewers will be asked to comment on the overlap between related submissions (http://journals.plos.org/plosone/s/submission-guidelines#loc-related-manuscripts).]

We will carefully review your manuscript upon resubmission and further consideration of the manuscript is dependent on the text overlap being addressed in full. Please ensure that your revision is thorough as failure to address the concerns to our satisfaction may result in your submission not being considered further.

3. In the online submission form, you indicated that [The data underlying the results presented in the study are available from the corresponding author on request.]. 

Reviewers' comments:

Reviewer's Responses to Questions

**Comments to the Author**

1. Is the manuscript technically sound, and do the data support the conclusions?

Reviewer #1: No

Reviewer #2: Partly

2. Has the statistical analysis been performed appropriately and rigorously? 

Reviewer #1: No

Reviewer #2: I Don't Know

3. Have the authors made all data underlying the findings in their manuscript fully available?

Reviewer #1: No

Reviewer #2: No

4. Is the manuscript presented in an intelligible fashion and written in standard English?

Reviewer #1: No

Reviewer #2: Yes

5. Review Comments to the Author

Reviewer #1: Influences of Atmospherics on Customer Satisfaction and Behavioral Intentions in the Restaurant Industry: Evidence from an Emerging Economy

I think the topic of this study is interesting in that it aims to identify the impacts of atmospherics of fine dining restaurant on customer satisfaction and behavioral intentions. However, the following issues must be addressed:

1. Upon a thorough examination of the entire paper, it becomes apparent that the study lacks a well-developed logical foundation based on theory. Despite collecting the data in New Delhi, the capital of India, there is a noticeable absence of a theoretical contribution. In other words, we cannot find a key theory that explains the theoretical background and contributions of this study compared to previous studies in the introduction and implications section.

2. Why did the authors use the term of configuration-related attributes? The meaning of the term is not clearly understood.

3. Since restaurant attributes consist of four sub-dimensions, the H1 must be presented for each subdimension.

4. Research Instrument and Data should be divided into 1) Instrument, 2) Sampling and data collection in the Methodology section.

5. The authors should make the Results section.

6. What is the difference between Table 4 and Table 5? To explain the measurement model, it may be sufficient to use just one appropriate table.

7. To verify the mediation effect, the authors only need to use one table, so please integrate the tables.

8. The authors need to update more recent papers.

9. Overall, the structure of the manuscript is not systematic, so please revise the structure of it.

10. Other

The references should be edited by the editorial policy of the journal.

Overall, the sentence is not concise, so please edit the entire sentence.

Please check the text carefully for grammar and typos.

Delete “Source: Authors’ own” below the tables and figures.

If the authors wish to show Figures of the AMOS analysis results, please correct them.

Reviewer #2: Underlying Theory is not mentioned.

Latest papers are not cited.

English grammar need to be improved.

More synthesis with the current literature will make the paper sound in the discussion section.

Table 10 not mentioned in the manuscript.

Future research areas not mentioned

6. PLOS authors have the option to publish the peer review history of their article (what does this mean? ). If published, this will include your full peer review and any attached files.

**Do you want your identity to be public for this peer review?** For information about this choice, including consent withdrawal, please see our Privacy Policy .

Reviewer #1: No

Reviewer #2: **Yes: ** Yogesh Devkinandan Mahajan

---

## [Author Response · Author response to Decision Letter 1]

4 Sep 2024

Influences of Atmospherics on Customer Satisfaction and Behavioral Intentions in the Restaurant Industry: Evidence from an Emerging Economy

Reviewer 01

I think the topic of this study is interesting in that it aims to identify the impacts of the atmospherics of fine dining restaurants on customer satisfaction and behavioural intentions. However, the following issues must be addressed:

1. Upon a thorough examination of the entire paper, it becomes apparent that the study lacks a well-developed logical foundation based on theory. Despite collecting the data in New Delhi, the capital of India, there is a noticeable absence of a theoretical contribution. In other words, we cannot find a key theory that explains the theoretical background and contributions of this study compared to previous studies in the introduction and implications section.

Action Taken: To investigate the relationship between the atmospherics of fine dining restaurants, customer satisfaction and behavioural intentions, we adopted the Stimulus-Organism-Response (SOR) theory proposed by Mehrabian and Russell (1974). The study aims to examine restaurant attributes as stimuli (S), overall customer satisfaction as an organism (O), and behavioural intentions as a response (R). Theoretical contributions have been addressed in the introduction and the implications section. (PP. 2,3, 23)

2. Why did the authors use the term of configuration-related attributes? The meaning of the term is not clearly understood.

Action Taken: The term Configuration-related attributes has been replaced by a more relevant term Spatial Configuration (Lima et al., 2024; Selem et al., 2023) to have a better understanding of its significance in fine dining restaurants. (PP. 5,7, supporting references have been added in the text as well as in the references section).

3. Since restaurant attributes consist of four sub-dimensions, the H1 must be presented for each subdimension.

Action Taken: HI is now being presented for all the four sub-dimensions. (Incorporated on PP. 8).

4. Research Instrument and Data should be divided into 1) Instrument, 2) Sampling and data collection in the Methodology section.

Action Taken: The suggestion has been incorporated and the methodology section has been divided into two Instrument & Data Collection. ((PP. 11,12).

5. The authors should make the Results section.

Action Taken: A separate section for results has been added. (PP. 19,20).

6. What is the difference between Table 4 and Table 5? To explain the measurement model, it may be sufficient to use just one appropriate table.

Action Taken: Table 5 has been removed now, and the numbering of the tables has now changed from Table 4 onwards. (PP. 16,17, Table 5 is now: Regression Weights Restaurant Atmospherics, Overall Customer Satisfaction and Behavioral Intentions).

7. To verify the mediation effect, the authors only need to use one table, so please integrate the tables.

Action Taken: Tables showing Direct & Indirect mediation effects have been integrated into one table and the Summary table for Mediation Analysis has also been removed to avoid duplicity. (PP. 18).

8. The authors need to update more recent papers.

Action Taken: Added Highlighted in yellow in the references section.

9. Overall, the structure of the manuscript is not systematic, so please revise the structure of it.

Action Taken: The structure of the manuscript has now been revised.

10. Other

The references should be edited according to the editorial policy of the journal.

Overall, the sentence is not concise, so please edit the entire sentence.

Please check the text carefully for grammar and typos.

Delete “Source: Authors’ own” below the tables and figures.

If the authors wish to show Figures of the AMOS analysis results, please correct them.

Action Taken: The above suggestions have been incorporated.

Reviewer 02

Reviewer Comment 01: The latest papers are not cited.

Action Taken: Thank you for your valuable feedback. We appreciate your suggestion to include the latest papers in our manuscript. In response to your comment, we have thoroughly reviewed recent literature and updated our manuscript accordingly.

Reviewer Comment: 02: English grammar needs to be improved.

Action Taken: Thank you for your insightful comment regarding the English grammar in our manuscript. To address this, we have thoroughly reviewed the text and made necessary corrections to enhance its clarity and readability. We also sought the assistance of a native English speaker to ensure that the language was polished and professional. These improvements have significantly enhanced the quality of our manuscript.

Reviewer Comment: 03: More synthesis with the current literature will make the paper sound in the discussion section.

Action Taken: Thank you for your valuable feedback on the need for more synthesis with current literature in the discussion section. We have carefully revised this section to integrate recent studies and provide a more comprehensive analysis.

Reviewer Comment: 04: Table 10 is not mentioned in the manuscript.

Action Taken: Thank you for pointing out the issue with Table 10. Upon review, we realized that Table 10 is not present in the manuscript and believe that the reference was intended for Table 1. We have now ensured that Table 1 is appropriately mentioned and discussed in the manuscript.

Reviewer Comment: 05: Future research areas not mentioned

Action Taken: Thank you for your insightful feedback regarding omitting future research areas. We agree that outlining future research directions is crucial for advancing the field. In response to your comment, we have added a section on future research areas in the conclusion of our manuscript.

---

## [Decision Letter · Decision Letter 1]

10 Sep 2024

PONE-D-24-10577R1Influences of Atmospherics on Customer Satisfaction and Behavioral Intentions in the Restaurant Industry: Evidence from an Emerging EconomyPLOS ONE

Dear Dr. Rathnasiri,

Thank you for submitting your manuscript to PLOS ONE. After careful consideration, we feel that it has merit but does not fully meet PLOS ONE’s publication criteria as it currently stands. Therefore, we invite you to submit a revised version of the manuscript that addresses the points raised during the review process.

We look forward to receiving your revised manuscript.

Kind regards,

Yasuko Kawahata

Academic Editor

PLOS ONE

Journal Requirements:

Reviewers' comments:

Reviewer's Responses to Questions

**Comments to the Author**

1. If the authors have adequately addressed your comments raised in a previous round of review and you feel that this manuscript is now acceptable for publication, you may indicate that here to bypass the “Comments to the Author” section, enter your conflict of interest statement in the “Confidential to Editor” section, and submit your "Accept" recommendation.

Reviewer #1: (No Response)

Reviewer #2: All comments have been addressed

2. Is the manuscript technically sound, and do the data support the conclusions?

Reviewer #1: Partly

Reviewer #2: Yes

3. Has the statistical analysis been performed appropriately and rigorously? 

Reviewer #1: No

Reviewer #2: I Don't Know

4. Have the authors made all data underlying the findings in their manuscript fully available?

Reviewer #1: Yes

Reviewer #2: Yes

5. Is the manuscript presented in an intelligible fashion and written in standard English?

Reviewer #1: No

Reviewer #2: Yes

6. Review Comments to the Author

Reviewer #1: This manuscript still has a lot to improve, even though the original comments have been revised to some extent. In particular, this study applied the SOR model, but it does not seem to have any theoretical contribution compared to previous studies.

The authors need to simplify several tables and present them as one, but they present too many tables unnecessarily, which leads to a lack of conciseness and clarity in the paper. This is also the case in the main text. Overall, the authors should write concisely.

What does *** mean in Table 4?

There are cases where the table presenting the results of the structural model analysis does not match the contents of the main text.

I do not know what the table explaining the parameters is trying to explain.

The authors should organize and present the results of the AMOS program analysis in Table 6 for the readers instead of presenting them as they are. What does *** mean in Table 6?

7.3. Future Directions -> 7.3. Limitations and Future Research Directions

The references should be edited by the editorial policy of the journal.

Figure 1 should display the four subdimensions of restaurant selection attributes.

I recommend the authors review other papers.

e.g.)

Jun, K., & Yoon, B. (2024). Consumer perspectives on restaurant sustainability: an SOR Model approach to affective and cognitive states. Journal of Foodservice Business Research, 1-24.

Reviewer #2: No comments. Paper is improved as per suggestions. No comments. Paper is improved as per suggestions.

7. PLOS authors have the option to publish the peer review history of their article (what does this mean? ). If published, this will include your full peer review and any attached files.

**Do you want your identity to be public for this peer review?** For information about this choice, including consent withdrawal, please see our Privacy Policy .

Reviewer #1: No

Reviewer #2: **Yes: ** Yogesh Mahajan

---

## [Author Response · Author response to Decision Letter 2]

23 Oct 2024

1. Theoretical contribution of the study - Theoretical contributions have been added on the page no.- 23

2. There are cases where the table presenting the results of the structural model analysis does not match the contents of the main text - Corrected on page no. 16,17 & 19

3. What does *** mean in Table 4? - p<0.01**inter-construct correlation value)

4. What does *** mean in Table 6? - p<0.001*** Inter-construct correlation value.

5. Organize and present the results of the AMOS program analysis in Table 6 for the readers instead of presenting them as they are - Incorporated on page no. 17 & 18

6. Limitations and Future Research Directions - Added on Page no. 23

7. Figure 1 should display the four subdimensions of restaurant selection attributes - Incorporated in the Figure 1

8. References should be edited by the editorial policy of the journal - It has been done

---

## [Decision Letter · Decision Letter 2]

1 Dec 2024

PONE-D-24-10577R2Influences of Atmospherics on Customer Satisfaction and Behavioral Intentions in the Restaurant Industry: Evidence from an Emerging EconomyPLOS ONE

Dear Dr. Rathnasiri, 

Thank you for submitting your manuscript to PLOS ONE. After careful consideration, we feel that it has merit but does not fully meet PLOS ONE’s publication criteria as it currently stands. Therefore, we invite you to submit a revised version of the manuscript that addresses the points raised during the review process.

We look forward to receiving your revised manuscript.

Kind regards,

Yasuko Kawahata

Academic Editor

PLOS ONE

Journal Requirements:

Reviewers' comments:

Reviewer's Responses to Questions

**Comments to the Author**

1. If the authors have adequately addressed your comments raised in a previous round of review and you feel that this manuscript is now acceptable for publication, you may indicate that here to bypass the “Comments to the Author” section, enter your conflict of interest statement in the “Confidential to Editor” section, and submit your "Accept" recommendation.

Reviewer #3: (No Response)

Reviewer #4: All comments have been addressed

2. Is the manuscript technically sound, and do the data support the conclusions?

Reviewer #3: Partly

Reviewer #4: Yes

3. Has the statistical analysis been performed appropriately and rigorously? 

Reviewer #3: Yes

Reviewer #4: Yes

4. Have the authors made all data underlying the findings in their manuscript fully available?

Reviewer #3: Yes

Reviewer #4: Yes

5. Is the manuscript presented in an intelligible fashion and written in standard English?

Reviewer #3: No

Reviewer #4: Yes

6. Review Comments to the Author

Reviewer #3: Thank you so much for giving me the opportunity to review this manuscript.

General Comments

The authors explored an interesting area of research, that is assessing the impact of atmospherics associated with customer satisfaction and behavioral intention of a fine dining restaurant. However, it is suggested to revised further to make it ready for publication.

Here are my comments and suggestions:

1. It is suggested to change the title as “Investigating the relationships of atmospherics, customer satisfaction and behavioural intentions: Evidence from the fine dining restaurant”.

2. “According to the National Restaurant Association of India, India Food Services Report, 2024, the Indian food industry is estimated to be Rs 5,69,487 crore for FY24. It is expected to grow to 7,76,511 crores by FY28, achieving a CAGR of 8.1 per cent, and it will grow with a CAGR of 13.2%”. It is suggested to convert these figures into US dollars.

3. P3. Third paragraph, what does “Divan” mean? Please consider global audience, when you discuss or introduce any terms.

4. It is suggested to club the literature review section and operational definition of constructs.

5. Table 5, relationships, it is suggested to reorder all the relationships (for example SA� CS, MENU� CS etc.).

6. It is suggested to merge the “Findings with “Discussions and Implications”.

7. In the discussions and implications section, it is suggested that theoretical implications must be first discussed in detail (which is too short), and then practical implications must be discussed, offering better insights.

8. It is suggested to strictly follow the format guidelines of the journal.

9. Proof reading is recommended.

Reviewer #4: 1. Kindly elaborate further on the research findings in your discussion.

2. Please refine your introduction by elaborating on the urgency of the factors and the significance of the topic.

7. PLOS authors have the option to publish the peer review history of their article (what does this mean? ). If published, this will include your full peer review and any attached files.

**Do you want your identity to be public for this peer review?** For information about this choice, including consent withdrawal, please see our Privacy Policy .

Reviewer #3: No

Reviewer #4: No

---

## [Author Response · Author response to Decision Letter 3]

30 Jan 2025

Reviewer 3 Comments & Revision

1. It is suggested to change the title as “Investigating the relationships of atmospherics, customer satisfaction andbehavioural intentions: Evidence from the fine dining restaurant” - The title has been changed on page no. 1

2. “According to the National Restaurant Association of India, India Food Services Report, 2024, the Indian food industryis estimated to be Rs 5,69,487 crore for FY24. It is expected to grow to 7,76,511 crores by FY28, achieving a CAGR of8.1 per cent, and it will grow with a CAGR of 13.2%”. It is suggested to convert these figures into US dollars - Incorporated on page no. 3

3. P3. Third paragraph, what does “Divan” mean? Please consider global audience, when you discuss or introduce anyterms - Incorporated on page no. 3

4. It is suggested to club the literature review section and operational definition of constructs - Incorporated on page no. 7, section 2.6

5. Table 5, relationships, it is suggested to reorder all the relationships (for example SA. CS, MENU. CS etc.). - Incorporated on page no. 17

6. It is suggested to merge the “Findings with “Discussions and Implications”. - Incorporated on page no. 20

7. In the discussions and implications section, it is suggested that theoretical implications must be first discussed in detail (which is too short), and then practical implications must be discussed, offering better insights. - Incorporated on page no. 22

8. It is suggested to strictly follow the format guidelines of the journal - Guidelines are now being followed

9. Proofreading is recommended - Proofreading has been done

Reviewer 4 Comment & Revision

1. Kindly elaborate further on the research findings in your discussion - Incorporated on page no. 20 & 21

2. Please refine your introduction by elaborating on the urgency of the factors and the significance of the topic - Incorporated on page no. 2

---

## [Decision Letter · Decision Letter 3]

11 Feb 2025

Influences of Atmospherics on Customer Satisfaction and Behavioural Intentions in the Restaurant Industry: Evidence from an Emerging Economy

PONE-D-24-10577R3

Dear Dr. Mananage Shanika Hansini Rathnasiri,

We’re pleased to inform you that your manuscript has been judged scientifically suitable for publication and will be formally accepted for publication once it meets all outstanding technical requirements.

Kind regards,

Yasuko Kawahata

Academic Editor

PLOS ONE

Additional Editor Comments (optional):

Reviewers' comments:

Reviewer's Responses to Questions

**Comments to the Author**

1. If the authors have adequately addressed your comments raised in a previous round of review and you feel that this manuscript is now acceptable for publication, you may indicate that here to bypass the “Comments to the Author” section, enter your conflict of interest statement in the “Confidential to Editor” section, and submit your "Accept" recommendation.

Reviewer #3: All comments have been addressed

Reviewer #4: All comments have been addressed

2. Is the manuscript technically sound, and do the data support the conclusions?

Reviewer #3: Yes

Reviewer #4: Partly

3. Has the statistical analysis been performed appropriately and rigorously? 

Reviewer #3: Yes

Reviewer #4: Yes

4. Have the authors made all data underlying the findings in their manuscript fully available?

Reviewer #3: Yes

Reviewer #4: Yes

5. Is the manuscript presented in an intelligible fashion and written in standard English?

Reviewer #3: Yes

Reviewer #4: Yes

6. Review Comments to the Author

Reviewer #3: The authors substantially revised the manuscript, based on reviewers' comments. It may be accepted for publication.

Reviewer #4: Its better if the research gap more clear in introduction. compare with previous research ..what a gap/novelty. the novelty is better if you make as highlight in the abstract

7. PLOS authors have the option to publish the peer review history of their article (what does this mean? ). If published, this will include your full peer review and any attached files.

**Do you want your identity to be public for this peer review?** For information about this choice, including consent withdrawal, please see our Privacy Policy .

Reviewer #3: No

Reviewer #4: No

---

## [Editor Report · Acceptance letter]

PONE-D-24-10577R3

PLOS ONE

Dear Dr. Rathnasiri,

I'm pleased to inform you that your manuscript has been deemed suitable for publication in PLOS ONE. Congratulations! Your manuscript is now being handed over to our production team.

Kind regards,

on behalf of

Dr. Yasuko Kawahata

Academic Editor

PLOS ONE